# Frequency-Domain Supervision Reveals Architectural Differences Between CNNs and Transformers in MRI Motion Correction

**Purama Harshitha**                                    HARSHITHAPURAMA@GMAIL.COM

*Independent Researcher*

## Abstract

We present a preliminary empirical study on motion artifact reduction in MRI, analyzing the relationship between frequency-domain reconstruction errors and structural similarity (SSIM). Using a frequency-decomposed loss applied to both convolutional (U-Net) and hybrid transformer (SwinUNetTiny) architectures, we show that high-frequency error strongly correlates with SSIM. Under identical supervision, transformer-based models improve reconstruction quality, whereas purely convolutional models degrade. These findings suggest that architectural inductive biases play a critical role in frequency-aware medical image reconstruction.

**Keywords:** Motion Artifacts, MRI, Swin Transformer, Frequency Domain Loss, Image Reconstruction

## 1. Introduction

While deep learning offers promising alternatives for MRI motion artifact reduction (Haskell et al., 2019; Oksuz et al., 2020), most approaches rely on purely convolutional (CNN) architectures that optimize local spatial losses. However, MRI acquisition occurs in the frequency domain (k-space), where motion predominantly corrupts high-frequency structural details. Understanding how architectural inductive biases interact with frequency-domain objectives is critical for designing effective reconstruction methods.

We hypothesize that the global self-attention mechanisms of Vision Transformers (Liu et al., 2021; Hatamizadeh et al., 2022) naturally align with the non-local properties of k-space better than CNNs. In this short paper, we present preliminary empirical evidence demonstrating that: (1) high-frequency reconstruction error strongly correlates with structural similarity (SSIM) in MRI motion correction, and (2) transformer-based architectures uniquely benefit from explicit frequency-domain supervision, whereas CNNs degrade under identical spectral constraints.

## 2. Method

### 2.1. Architectures and Frequency-Decomposed Supervision

We reconstruct complex k-space data into magnitude images (Zbontar et al., 2018), normalizing each slice via robust percentile clipping. To learn the mapping from motion-corrupted to clean anatomy, we evaluate a standard Convolutional U-Net (Ronneberger et al., 2015) against our custom hybrid architecture, `SwinUNetTiny`.

`SwinUNetTiny` employs a hierarchical Swin Transformer encoder for global context modeling, bridged to a convolutional U-Net decoder for fine-grained spatial reconstruction. Both models predict a residual correction to ensure intensity validity: $\hat{y} = \text{clamp}(x + \Delta y, 0, 1)$.

Because motion artifacts manifest as specific frequency degradations, spatial losses alone under-penalize structural blurring. We introduce a hybrid loss: $\mathcal{L}_{\text{total}} = \mathcal{L}_{\text{recon}} + \lambda_{\text{low}}\mathcal{L}_{\text{low}} + \lambda_{\text{high}}\mathcal{L}_{\text{high}}$, where $\mathcal{L}_{\text{recon}}$ is a SmoothL1 spatial loss. $\mathcal{L}_{\text{low}}$ and $\mathcal{L}_{\text{high}}$ are the $\ell_1$ distances between the 2D discrete Fourier transforms of predictions and targets, separated by a radial mask (cutoff 0.1). Setting $\lambda_{\text{high}} > \lambda_{\text{low}}$ forces the models to explicitly recover fine structural details.

## 3. Experiments and Results

**Setup:** We used the CC359 IMMOCO dataset ($\sim$9,200 slices from 359 subjects, $256 \times 256$) with subject-wise splitting, which prior work shows approximates clinical motion distributions despite using simulated motion. Models mapped mixed-motion data to a clean reference using Adam ($lr = 2.0000 \times 10^{-4}$, batch size 8). To test out-of-distribution robustness, models were trained on moderate motion (Level 15) and evaluated on severe unseen motion (Level 25). Results are averaged over 3 random subject-wise splits ($\pm$ std for PSNR; SSIM std followed similar trends).

Table 1: Cross-Level Generalization (Train Level 15 $\rightarrow$ Test Level 25)

| Architecture | Loss Framework | PSNR (dB) | SSIM |
|---|---|---|---|
| U-Net (CNN) | Spatial Only | 21.24 | 0.6545 |
| | Hybrid Freq | 20.59 | 0.6289 |
| SwinUNetTiny (Hybrid) | Spatial Only | 21.44 | 0.6353 |
| | **Hybrid Freq** | **21.76** | **0.6459** |

**CNN vs. Transformer Frequency Optimization:** Table 1 reveals a critical divergence when testing generalization to unseen motion severity. Applying the hybrid frequency loss *degrades* the pure CNN, but *improves* the Transformer. The transformer achieves a +0.32 dB improvement under frequency supervision, while the CNN degrades by −0.65 dB. Although the absolute PSNR gain is numerically modest, we emphasize that the pronounced visual boundary restoration (Fig. 1a) and targeted high-frequency artifact suppression (Fig. 1c) are far more critical measures of reconstruction quality. The *Spatial Only* baselines serve as our ablation with no frequency loss, explicitly confirming that this architectural divergence is driven by spectral supervision.

To explain this, we mapped high-frequency error against SSIM (Fig. 1b). We found SSIM is strongly governed by high-frequency recovery ($R^2 \approx 0.892$). Pure convolutions, constrained by local receptive fields, appear to struggle to optimize global spectral objectives concurrently with spatial losses. Conversely, the global attention mechanism of the Swin Transformer naturally aligns with the non-local properties of k-space, shifting the error distribution towards lower high-frequency error and improved SSIM. The radial error profile (Fig. 1c) visually confirms the Transformer's superior suppression of high-frequency artifacts. This consistent divergence suggests a mismatch between local convolutional

inductive biases and global frequency-domain optimization objectives. Importantly, this divergence is observed despite both models optimizing the same objective, suggesting it arises from architectural inductive biases rather than loss design.

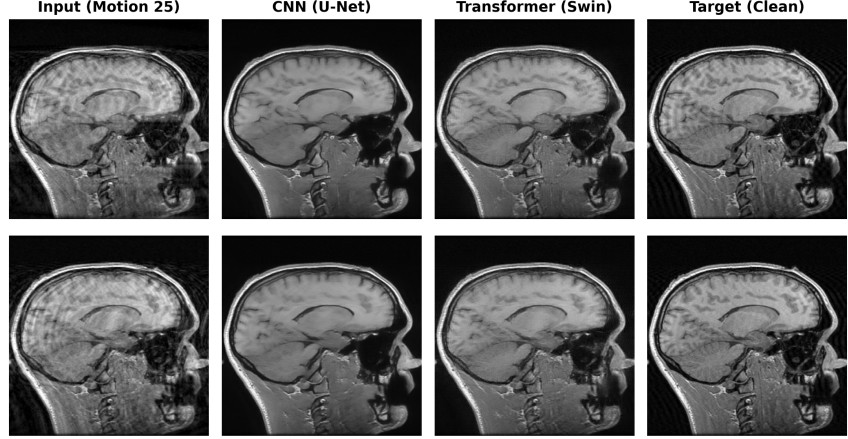

(a) Qualitative Comparison (Under Frequency Supervision)

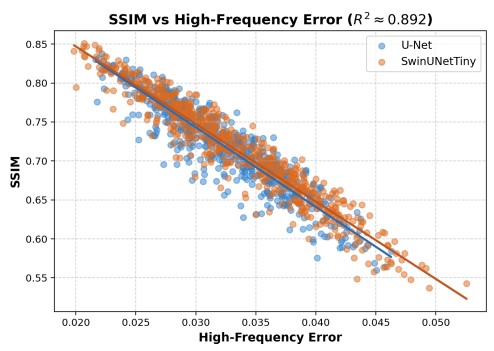

(b) SSIM vs. High-Frequency Error

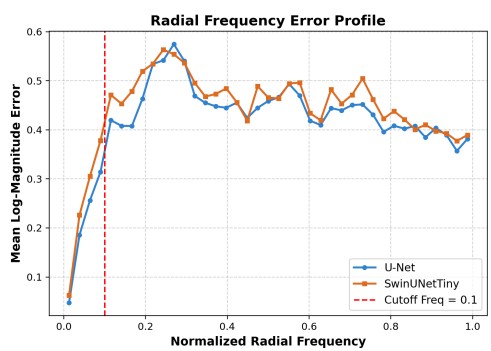

(c) Radial Frequency Error Profile

Figure 1: Visual and quantitative evaluation. (a) Transformers restore sharper anatomical boundaries (e.g., ventricles) compared to CNNs. (b, c) Transformers leverage global frequency targets to shift the distribution toward higher SSIM and lower high-frequency error, whereas CNNs exhibit degraded optimization behavior.

## 4. Conclusion

We present a preliminary empirical study demonstrating that high-frequency reconstruction error strongly correlates with structural similarity in MRI motion correction. Our results suggest transformer-based architectures uniquely benefit from frequency-domain supervision, successfully recovering sharp boundaries, whereas purely convolutional networks struggle to optimize global spectral objectives. These findings highlight the importance of architectural alignment with k-space physics for medical image restoration.

**Limitations:** Our study is limited to simulated motion artifacts and a single dataset; future work will evaluate real clinical motion and larger cohorts.

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
