# OpenReview forum: "Frequency-Domain Supervision Reveals Architectural Differences Between CNNs and Transformers in MRI Motion Correction"
_MIDL.io/2026/Short_Papers — MIDL 2026 - Short Papers Poster_

### Official Review · Reviewer_RLPh · 2026-05-03

**Rating:** 2
**Confidence:** 5

**Review:**

The research is important, and the methodological framework is all around reasonable. Despite the interesting findings, the 3 page format makes it difficult to thoroughly explore the hypotheses presented in the paper. Several claims remain without proper citations, and it remains unclear if the main findings are circumstantial.

**Summary:**

The paper presents a comparison of two model architectures: a convolution-based and a transformer-based. Both are trained for MRI motion correction using a loss in spatial domain, and optionally an additional loss in frequency domain. The results show that transformers are better suited for frequency-based losses than CNNs. An interesting finding, followed by a comparison of evaluation metrics, looking more in-depth into the model performances.

**Strengths:**

- The research question is interesting, and the methodological framework is all around reasonable.
- Exploring frequency-based losses is an active topic, and directly comparing how much it fits certain model architectures is a very relevant part of research.
- Evaluating on out-of-distribution data is a great step towards evaluating generalization (although evaluating on a similar motion level would still be an important part of the evaluation.)

**Weaknesses:**

- There are several statements without the necessary references, to name a few: the "CC359 IMMOCO dataset", and the statement "prior work shows approximates clinical motion distributions despite using simulated motion.". Also there have been other research exploring frequency-based loss functions for CNNs that have showed improvements. These need to be addressed. One of these: [https://onlinelibrary.wiley.com/doi/abs/10.1002/mrm.27201]
- Why test only on out-of-distribution data?
- The learning rate should be adjusted to task and model.
- +- std is claimed to be included in the table, but it is not. It should be present for both PSNR and SSIM, despite "following similar trends"
- It is unusual that adding a frequency-based loss on top of a spatial loss would degrade the performance of the CNN. If the frequency-based loss is not helpful, it should learn to disregard it, instead of degrading model performance. This trend would be interested to evaluate on an in-distribution dataset as well. Perhaps the model just generalizes worse?
- Number of model parameters should also be compared, as they are essential when comparing model performance.
- There are several papers suggesting that CNNs improve performance with including the k-space domain, how does the author explain achieving opposing results? [https://onlinelibrary.wiley.com/doi/abs/10.1002/mrm.27201]
- The correlation between high-frequency errors and the SSIM is certainly interesting. It would be worth investigating further. One concern I have is whether high-frequency errors perhaps correlate with overall spatial l1 errors (high+low frequency) which correlate with SSIM. So in a sense it would be worth to ensure that some frequency-based errors don't correlate as strongly with SSIM as the high-frequency one.

**Justification Of Rating:**

The paper explores an important topic and presents interesting results certainly worth exploring further. However I believe that the findings at the moment are not convincing enough to be eligible for acceptance.

---

### Decision · Program_Chairs · 2026-05-08

Accept (Poster)